# Estimating Transfer Entropy under Long Ranged Dependencies

**Sahil Garg**[1,*]   **Umang Gupta**[2]   **Yu Chen**[1,†]   **Syamantak Datta Gupta**[1,†]   **Yeshaya Adler**[1,†]   **Anderson Schneider**[1]

**Yuriy Nevmyvaka**[1]

[1]Department of Machine Learning Research, Morgan Stanley, New York, NY, USA,
[2]Department of Computer Science, University of Southern California, Los Angeles, CA, USA
[*]Corresponding Author: sahil.garg.cs@gmail.com, sahil.garg@morganstanley.com
[†]Equal Contributions

## Abstract

Estimating Transfer Entropy (TE) between time series is a highly impactful problem in fields such as finance and neuroscience. The well-known nearest neighbor estimator of TE potentially fails if temporal dependencies are *noisy and long ranged*, primarily because it estimates TE *indirectly* relying on the estimation of *joint entropy* terms in high dimensions, which is a hard problem in itself. Other estimators, such as those based on Copula entropy or conditional mutual information have similar limitations. Leveraging the successes of modern discriminative models that operate in high dimensional (noisy) feature spaces, we express TE as a difference of two *conditional entropy* terms, which we *directly* estimate from conditional likelihoods computed in-sample from any discriminator (timeseries forecaster) trained per maximum likelihood principle. To ensure that the in-sample log likelihood estimates are not overfit to the data, we propose a novel perturbation model based on locality sensitive hash (LSH) functions, which regularizes a discriminative model to have smooth functional outputs within local neighborhoods of the input space. Our estimator is consistent, and its variance reduces linearly in sample size. We also demonstrate its superiority w.r.t. state-of-the-art estimators through empirical evaluations on a synthetic as well as real world datasets from the neuroscience and finance domains.

## 1 INTRODUCTION

Information theory plays a central role in modern machine learning for tasks like clustering, feature selection, representation learning, autoencoding, generative modeling, fairness, etc. [Shannnon, 1948, Cover, 1999, Cicalese et al., 2019,

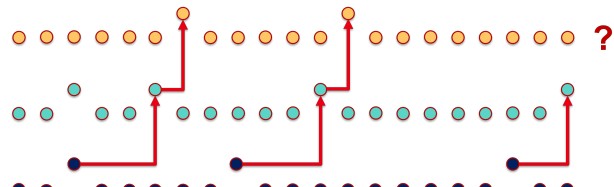

Figure 1: Three different time series are shown in order to illustrate *transfer entropy* (TE). There is a clear pattern of TE from the second time series (green) to the first one (yellow), i.e. predictability of the observations in the first time series given the knowledge of the second one. Similarly, there is TE from the third time series (navy) to the second one, though the dependencies are relatively complex.

Kingma and Welling, 2019, Song et al., 2019, Song and Kingma, 2021]. A relatively new concept in information theory, introduced by Schreiber [2000], is *transfer entropy* (TE) that quantifies the reduction in uncertainty about one time series given another (see Fig. 1). TE is theoretically and practically appealing for various domains, including finance and neuroscience [Vicente et al., 2011, Jizba et al., 2012, Ver Steeg and Galstyan, 2012, Ursino et al., 2020, Restrepo et al., 2020, Sipahi and Porfiri, 2020].

With the recent rapid advance of simultaneous high-density recordings of neural activities across multiple brain areas [Siegle et al., 2021, Steinmetz et al., 2021], it is essential to have a scalable and robust model for estimating TE between neural ensembles in the presence of sparse signal and large noise due to ubiquitous neuron-to-neuron or trial-to-trial variance [Steinmetz et al., 2018, Kass et al., 2018]. In the finance domain, given the low-signal-to-noise ratio, empirical models leverage advances in TE estimation by filtering out very weak explanatory time series [Dimpfl and Peter, 2013, Sandoval, 2014].

In practice, it is challenging to estimate TE especially under *long-ranged & noisy temporal dependencies* [Lindner et al., 2011, Barnett and Bossomaier, 2012, Zhang et al.,

*Accepted for the 38th Conference on Uncertainty in Artificial Intelligence* (UAI 2022).

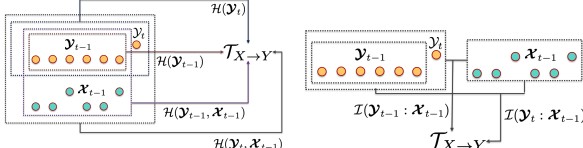

(a) kNN or Copula Estimator.  (b) ITENE Estimator.

Figure 2: Limitations of Estimators. The yellow dots represent the past ($\mathcal{Y}_{t-1}$) and the present ($\mathcal{Y}_t$) of timeseries Y and the green ones are for the past ($\mathcal{X}_{t-1}$) of timeseries X. Present observation for timestep $t$ has noisy dependencies w.r.t. its own past and of X. In 2(a), kNN based estimation of TE requires estimation of four joint entropy terms in high dimensions which is a harder problem, unwarranted, and susceptible to dependency based noise. Same limitation applies to ITENE in 2(b), as it computes TE as a difference of two mutual information terms in high dimensions.

2019]. A popular technique for estimating TE is based on the $k$ nearest neighbors (kNN) method [Kraskov et al., 2004, Lindner et al., 2011]. However, this measures TE indirectly, through joint entropy (Kozachenko and Leonenko [1987], Singh and Póczos [2016]). This becomes problematic when temporal dependencies are noisy. A related approach is to estimate TE via Copula joint entropy or conditional mutual information [Ma, 2019, Zhang et al., 2019], which is also susceptible to dependency based noise. See Fig. 2.

Noting that TE is represented by two conditional entropy components, in this paper, we propose a discriminative learning approach to the problem. Specifically, we obtain *in-sample* estimates of conditional likelihoods from a discriminative model to estimate the conditional entropy terms directly. This allows us to exploit modern machine learning methods to predict a low dimensional variable, conditioned on a very high dimensional variable, even when many of the dimensions are noisy. Any discriminative model, *trained as per the maximum likelihood principle*, can obtain *in-sample* estimates of the conditional log likelihood.

For instance, deep neural nets trained with mean squared error as the loss function provide estimates of the conditional log likelihood for free, if errors are assumed to be Gaussian distributed. In our approach, one can also employ any probabilistic regression model for obtaining the conditional likelihoods [Dabney et al., 2018, Alexandrov et al., 2020, Guen and Thome, 2020, Rasul et al., 2021, Gouttes et al., 2021, Tang and Matteson, 2021, Pal et al., 2021, Yoon et al., 2022, Das et al., 2022]. Though we advocate for the simpler approach mentioned above as it accommodates a large variety of time series forecasters [Bai et al., 2018, Oreshkin et al., 2019, Kitaev et al., 2019, Benidis et al., 2020, Zeng et al., 2021, Fan et al., 2021, Gu et al., 2021, Challu et al., 2022]. For discrete-valued time series, one simple and generic choice is to obtain conditional log likelihood from any classifier trained with cross-entropy loss.

We must ensure that the in-sample estimates of conditional log likelihood are not overfit to the time series. This is particularly relevant when the estimate is for quantifying whether additional information from another time series improves the (in-sample) predictability of a given time series. Intuitively, a highly expressive discriminator (timeseries forecaster) tends to learn a non-smooth function overfitting w.r.t. training data points sparsely populated in high-dimensional space. For this aspect, we take inspiration from the literature on adversarial learning for deep neural nets, where high susceptibility of models to small noise, imperceptible to humans, is a common problem. Such a phenomenon is observed despite standard regularization techniques, such as weight decay, dropout, batch normalization, etc. Quoting Yoshida and Miyato [2017], "adversarial training is designed to achieve insensitivity to the perturbation of training data" [Goodfellow et al., 2015, Zhao et al., 2020, Dong et al., 2020]. While for the problem of TE estimation, there is neither an "adversary", nor a need to generalize to unseen domains [Volpi et al., 2018], the *important takeaway* is that the functional outputs of an expressive discriminator must be regularized to be consistent w.r.t. the perturbations of its input, thus ensuring local Lipschitz-like properties of the output function [Yang et al., 2020, Jiang et al., 2020]. This allows a safe and robust in-sample estimation of TE.

The challenge is to select an appropriate perturbation model. Adding Gaussian noise is a popular choice. We argue in favor of an even more general perturbation model, which may be agnostic to the data distribution while respecting the underlying data manifold locally (since the desired smoothness of model outputs is *local* in input space), and preferably non-stationary w.r.t. input space. Accordingly, we propose a novel perturbation model, based on *locality sensitive hashing (LSH)*, which is a well known technique for finding nearest neighbors in high dimensions [Indyk and Motwani, 1998, Kulis and Grauman, 2009, Grauman and Fergus, 2013, Zhao et al., 2014, Wang et al., 2017]. As per the theory of LSH, a hashcode represents a local neighborhood in the input space, characterizing the underlying data manifold locally. We propose that the outputs of a discriminative model should be consistent within a hashcode bin (one can think of it as a histogram bin in high dimensions), which is accomplished by generating perturbations local to a bin.

Sampling perturbations from a hashcode bin doesn't introduce any bias since the hashcode bins correspond to data-driven histograms capable of characterizing the underlying true data distribution [Lugosi and Nobel, 1996] as we show in our theoretical analysis of the estimator (Sec. 3.2). For practical purposes, perturbations are generated from a convex combination of the existing data points from the same bin (i.e. sampling from within a convex hull of datapoints); see Fig. 3. Furthermore, we define a simple yet effective information theoretic measure to ensure the consistency of the model outputs, i.e. *minimize conditional entropy* of

the model output given the locality sensitive hashcodes of perturbed inputs.

The rest of this paper is organized as follows. After discussing the basics of TE and related works in Sec. 2, we present a novel TE estimator in Sec. 3, along with theoretical guaranties (Sec. 3.2). A thorough empirical analysis using a synthetic dataset, a neuroscience dataset of activity in different brain regions, and two financial datasets of high frequency trading activity in US stocks is provided in Sec. 4. Code is availed here: `github.com/morganstanley/MSML/papers/Direct_Estimate_Transfer_Entropy`.

## 2 BACKGROUND

Transfer entropy, introduced originally by Schreiber [2000], refers to the reduction in uncertainty for forecasting a time series given the knowledge of another time series.

Let $X$, $Y$ be two discrete- or real-valued time series. Let $\mathcal{X}_t$, $\mathcal{Y}_t$ be the random variables denoting $X$, $Y$ at time $t$ respectively, where $t \in \{0, 1, \ldots, \}$, and let $x_t$, $y_t$ be their observed values. Furthermore, let $\boldsymbol{\mathcal{Y}}_t$ denote the $t$-dimensional vector $\boldsymbol{\mathcal{Y}_t} \equiv (\mathcal{Y}_0, \cdots, \mathcal{Y}_t)$; and let $\mathbf{y}_t$ be a realization of $\boldsymbol{\mathcal{Y}}_t$. Then, the conditional entropy of $\mathcal{Y}_t$, given its past observations, i.e., the uncertainty in forecasting $Y$ for the current timestep, conditioned on its history, is represented as

$$\mathcal{H}(\mathcal{Y}_t|\boldsymbol{\mathcal{Y}}_{t-1}) \equiv \mathbb{E}_{\mathbf{y}_{t-1} \sim \boldsymbol{\mathcal{y}}_{t-1}} \left[ \log p(y_t|\mathbf{y}_{t-1}) \right]. \quad (1)$$

Uncertainty when forecasting time series Y, given the knowledge of both its own past as well as X's previous realizations, is expressed as the following conditional entropy: $\mathcal{H}(\mathcal{Y}_t|\boldsymbol{\mathcal{Y}}_{t-1}, \boldsymbol{\mathcal{X}}_{t-1})$. Mathematically, TE is the difference between the two conditional entropy terms, measuring the additional information on $\mathcal{Y}_t$ available in the past realizations of $X$, that is not already present in the past of $Y$.

$$\mathcal{T}_{X \to Y} = \mathcal{H}(\mathcal{Y}_t|\boldsymbol{\mathcal{Y}}_{t-1}) - \mathcal{H}(\mathcal{Y}_t|\boldsymbol{\mathcal{Y}}_{t-1}, \boldsymbol{\mathcal{X}}_{t-1}) \geq 0 \quad (2)$$

A popular approach for estimating TE is based on $k$ nearest neighbors [Lindner et al., 2011, Zhu et al., 2015], which measures TE indirectly through joint entropy [Kozachenko and Leonenko, 1987], so Eq. 2 must be re-written as,

$$\mathcal{T}_{X \to Y} = \mathcal{H}(\mathcal{Y}_t|\boldsymbol{\mathcal{Y}}_{t-1}) - \mathcal{H}(\mathcal{Y}_t|\boldsymbol{\mathcal{Y}}_{t-1}, \boldsymbol{\mathcal{X}}_{t-1})$$
$$= \mathcal{H}(\boldsymbol{\mathcal{Y}}_t) - \mathcal{H}(\boldsymbol{\mathcal{Y}}_{t-1}) - \mathcal{H}(\boldsymbol{\mathcal{Y}}_t, \boldsymbol{\mathcal{X}}_{t-1}) + \mathcal{H}(\boldsymbol{\mathcal{Y}}_{t-1}, \boldsymbol{\mathcal{X}}_{t-1}).$$

Due to significantly different scales of distances across these four terms, the error biases do not cancel each other. Attempts to correct the compounding of biases by estimating joint entropy terms together, using nearest neighbors in the joint space of all the variables, cannot adequately address vulnerability to the dependency based noise [Kraskov et al., 2004], Lindner et al. [2011]. The above formulation is particularly problematic when $Y$ has a long memory, and consequently, the conditional random variable, $\boldsymbol{\mathcal{Y}}_{t-1}$, is high dimensional, with noisy dependencies w.r.t. the target variable

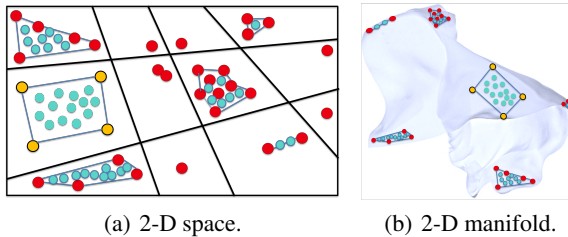

(a) 2-D space.      (b) 2-D manifold.

Figure 3: This figure illustrates the concept of hashcode-based perturbations to regularize the output function. In 3(a), data points (*Red* or *Yellow* dots) are dispersed in 2-D space; the lines represent hash functions, and their intersections correspond to *hashcode bins*. Each bin represents a local neighborhood in the input space, data points in a bin being neighbors of each other. Owing to the locality of a hashcode bin w.r.t. the data manifold, perturbations are generated (*Blue* dots) in a bin from randomly sampled convex combinations of the existing data points. We propose that model outputs for the perturbations within a hashcode bin should be consistent w.r.t. each other. In 3(b), hashcode bins from 3(a) are shown on a manifold embedded in 3-D space, illustrating how hashcode bins represent the data manifold locally, thereby leading to a non-stationary perturbation function. For instance, the bin corresponding to yellow dots has a data distribution of highest entropy, thus implying perturbations of the greatest magnitude in the bin.

$\mathcal{Y}_t$; for example, when only a few of the many dimensions in $\boldsymbol{\mathcal{Y}}_{t-1}$ explain $\mathcal{Y}_t$, while the other ones are noise components. Same applies for noisy dependencies of $\boldsymbol{\mathcal{X}}_{t-1}$ w.r.t. $\mathcal{Y}_t$, when estimating $\mathcal{H}(\mathcal{Y}_t|\boldsymbol{\mathcal{Y}}_{t-1}, \boldsymbol{\mathcal{X}}_{t-1})$. When estimating TE between time series from real world domains like finance and neuroscience, it is natural to expect such long ranged, noisy temporal dependencies. In a recent work, Ma [2019] shows TE to be equivalent to Copula entropy, but their estimator also relies upon estimating joint entropy terms in high dimensions. Kernel density or histogram-based estimators [Ver Steeg and Galstyan, 2012, Zuo et al., 2013] also suffer from dependency based noise. Even without noise, these estimators are efficient only in low dimensions.

TE can also be expressed as conditional mutual information:

$$\mathcal{T}_{X \to Y} = \mathcal{I}(\boldsymbol{\mathcal{X}}_{t-1} : \mathcal{Y}_t) - \mathcal{I}(\boldsymbol{\mathcal{X}}_{t-1} : \boldsymbol{\mathcal{Y}}_{t-1}). \quad (3)$$

While theoretically appealing, it is notoriously difficult to estimate mutual information between two high dimensional variables. Zhang et al. [2019] propose to estimate TE through Eq. 3 using the deep neural nets based estimator of mutual information (MINE) due to Belghazi et al. [2018]. McAllester and Stratos [2020] show that MINE has high variance which increases with true mutual information value itself, which renders it unsuitable for estimating TE. Next, we introduce a novel estimator of TE in Sec. 3.

# 3 DIRECT ESTIMATE OF TE

We propose a direct empirical estimate of TE leveraging upon highly expressive timeseries forecasting models such as in deep learning. While the key idea is simple and general, we discuss how and why this approach should work well in practice despite potential concerns such as model mis-specification, mis-calibration, etc. Moreover, we introduce a novel perturbation based regularization model for ensuring a robust estimate of TE. In Sec. 3.2, we establish that the estimator is consistent with low variance.

Referring back to Eq. 2 in Sec. 2, we propose to estimate TE via the direct estimation of the *conditional entropy* terms. $\mathcal{H}(\mathcal{Y}_t|\boldsymbol{\mathcal{Y}}_{t-1})$ is expressed as,

$$\hat{\mathcal{H}}(\mathcal{Y}_t|\boldsymbol{\mathcal{Y}}_{t-1}) = -\frac{1}{n}\sum_{i=1}^{n}\log p(y_t^{(i)}|\mathbf{y}_{t-1}^{(i)}). \tag{4}$$

Identical estimation logic applies for $\mathcal{H}(\mathcal{Y}_t|\boldsymbol{\mathcal{Y}}_{t-1}, \boldsymbol{\mathcal{X}}_{t-1})$. Here, we make an important observation: a discriminative model that is trained by *maximizing the conditional log likelihood* of the target variable given the input variable can be employed as an estimator of conditional entropy.

For discrete-valued time series with support set $Z$, a classifier trained by cross entropy loss can provide an empirical estimate of conditional entropy itself.

$$\hat{\mathcal{H}}_q(\mathcal{Y}_t|\boldsymbol{\mathcal{Y}}_{t-1}) = \frac{1}{n}\sum_{i=1}^{n} -\sum_{z\in Z}\mathbb{I}_{y_t^{(i)}=z}\log f_z(\mathbf{y_{t-1}^{(i)}}) \tag{5}$$

$$= \frac{1}{n}\sum_{i=1}^{n} -\log q(y_t^{(i)}|\mathbf{y}_{t-1}^{(i)}), \tag{6}$$

where $\mathbb{I}(.)$ is an indicator function; $f(\mathbf{y}_{t-1}^{(i)})$ is output of the classifier, a multinomial vector of inferred class probabilities; $\log q(y_t^{(i)}|\mathbf{y}_{t-1}^{(i)})$ is an estimate of conditional log likelihood. By Jensen's inequality, it is well known that a proposal distribution $q(.)$ for $p(.)$ upper bounds the corresponding entropy function, $\mathcal{H}_q \geq \mathcal{H}_p$, with the error bias being KL-divergence between the two distributions, $\mathcal{D}_{KL}(p(.|.)||q(.|.)) > 0$ [Cover, 1999].

For continuous valued time series, any regression model trained with mean squared error as the loss function can be employed to estimate conditional entropy if errors are assumed to be Gaussian distributed: $y_t \sim \mathcal{N}(f(\mathbf{y}_{t-1}), \sigma)$. $f(\mathbf{y}_{t-1})$ can be a highly expressive deep neural net which can essentially approximate any functional relationship between target $y_t$ and input $\mathbf{y}_{t-1}$.

Besides the above two simple and generic choices which are prevalent in literature of supervised deep learning, any deterministic discriminator (timeseries forecaster) trained with maximum likelihood objective or a probabilistic discriminator is equally applicable here.

As per the above, a discriminator can act as a conditional entropy estimator with the advantage of being efficient even in very high dimensional noisy feature spaces. The empirical estimate of TE is directly expressible in terms of the ratio of the two conditional likelihood terms, $q(y_t|\mathbf{y}_{t-1}, \mathbf{x}_{t-1}))$ and $q(y_t|\mathbf{y}_{t-1})$. The estimator has an *error bias* naturally inherited by the model bias in the discriminator. The choice of model including its hyperparameters, $q(.|.)$, is the same for estimating the two terms. This should help reducing the model bias since the empirical estimate relies only on the ratio of the two terms.

Furthermore, since the goal is to quantify the decrease in uncertainty and not to maximize the accuracy of the original model for forecasting, from many choices of neural architectures for timeseries forecasting such as TCNs, RNNs, Transformers, etc., one which is known to be more robust to overfitting, model mis-calibration, is preferred. One can even use more lightweight versions of neural architectures than those used for forecasting, and employ standard generalization techniques such as weight decay, dropout, early stopping, small batch size, etc.

In theory, the error bias for estimating TE with conditional likelihood $q(.|.)$ stemming from a discriminator, as opposed to true conditional likelihood $p(.|.)$, is as below.

$$\mathcal{T}_{X\to Y}^q - \mathcal{T}_{X\to Y} = \mathcal{D}_{KL}(p(y_t|\mathbf{y}_{t-1})||q(y_t|\mathbf{y}_{t-1}))$$
$$-\mathcal{D}_{KL}(p(y_t|\mathbf{y}_{t-1}, \mathbf{x}_{t-1})||q(y_t|\mathbf{y}_{t-1}, \mathbf{x}_{t-1})) \tag{7}$$

The two terms in the r.h.s. are KL-divergence terms which are error biases of $\mathcal{H}^q(\mathcal{Y}_t|\boldsymbol{\mathcal{Y}}_{t-1})$ and $\mathcal{H}^q(\mathcal{Y}_t|\boldsymbol{\mathcal{Y}}_{t-1}, \boldsymbol{\mathcal{X}}_{t-1})$ respectively. Since both terms are non-negative, $\mathcal{D}(.||.)) \geq 0$, they *counteract* each other leading to a smaller magnitude of the overall error bias of TE. Theoretically, one should expect the error bias of $\mathcal{H}^q(\mathcal{Y}_t|\boldsymbol{\mathcal{Y}}_{t-1}, \boldsymbol{\mathcal{X}}_{t-1})$ to be larger than or equal to its counterpart due to the conditioning upon a higher number of dimensions; this should lead to a net negative error bias. In addition, there is also a bias due to the finite sample size for both of the conditional entropy terms, which could be positive or negative; we analyze variance of the estimates due to finite samples size in Sec. 3.2.

For practical purposes, we suggest normalizing TE with the first conditional entropy term in Eq. 2, quantifying relative decrease in uncertainty. This measure is more robust to a potential issue of model mis-calibration.

For ensuring a robust estimate of TE, we also propose a perturbations based regularizer.

## 3.1 REGULARIZE VIA LSH-PERTURBATIONS

Machine learning literature provides a plethora of discriminative models for time series modeling which can be employed in our TE estimator. The challenge, however, is that an expressive discriminator like a deep neural net, with an ability to learn any function, can overfit to training data, even

under standard generalization techniques such as weight decay, dropout, etc. In sparsely populated regions of the underlying manifold of model inputs, the output function may be non-smooth.

Inspired by recent works in adversarial training of deep neural nets [Goodfellow et al., 2015, Zhao et al., 2020, Dong et al., 2020], we propose to accomplish Lipschitz-like smoothness of the output function by ensuring that model outputs are consistent w.r.t. perturbations in inputs. A good choice for a perturbation model should be able to characterize the underlying data manifold locally, since the perturbations are supposed to be local w.r.t. inputs.

The general concept of data perturbations based regularization can be formalized as below.

$$\bar{\mathbf{y}} \sim g(\mathbf{y}) \tag{8}$$

Here, $g(.)$ is a model that generates perturbations for a given input $\mathbf{y}$. An explicit way to ensure that the model output function, $f(.)$, is consistent w.r.t. perturbations in input, i.e., $f(\mathbf{y}) = f(\bar{\mathbf{y}})$, is to define a regularization penalty for inconsistent model outputs on perturbations. Later in this section, we present an information theoretic regularization criterion which expects model outputs on perturbations of a given input to be of low entropy. One can even employ a non-invasive regularization by tuning the hyper-parameters of a model. An implicit way to ensure consistency is to augment the training data with perturbed data points. Data augmentation is not used here to generalize to unseen domains, but to ensure that the learned model output function is smooth in the local vicinity of training data points, especially if those points were sparsely populated in the input space.

A perturbation model challenges the primary (discriminative) model by perturbing its inputs locally in the data manifold. It can be parametric or non-parametric: perturbations based on Gaussian noise are described in [Rothfuss et al., 2019, Maaten et al., 2013, Bishop, 1995]. A good choice of perturbation model should characterize the data manifold locally, whereas the primary model may be inefficient at modeling the local manifold. While not necessary for practical purposes, if the perturbation model can also characterize the underlying true data distribution, $p(\mathbf{y}_{t-1}, y_t)$, it would theoretically ensure that there is no error bias from using the perturbations based regularization or data augmentations for estimating TE as we show in Sec. 3.2.

We propose a perturbation model based upon *locality sensitive hashing* (LSH), which perturbs inputs in local neighborhoods of input space. Such neighborhoods are defined from the hashcodes that split the input space into different regions. LSH is a randomized algorithm, that is proven to be efficient in finding nearest neighbors in very high dimensions [Indyk and Motwani, 1998, Zhao et al., 2014, Wang et al., 2017]. The core idea is that similar data points according to some distance metric are assigned the same hashcodes with probability inversely proportional to the distance metric.

This theoretical property of hashcodes implies that a hashcode bin represents a local neighborhood (manifold) in input space. We propose a hashcode-based regularization such that model outputs are smooth w.r.t. perturbations of inputs within a hashcode bin. We generate perturbations from randomly sampled *convex combinations* of the existing data points in a bin; see Fig. 3 for an illustration. In essence, LSH plays the role of histograms in high dimensions. In histogram bins, one has explicit boundaries of bins, so sampling from within a bin is easier. Whereas in a high dimensional setting, the boundaries of a bin can only be estimated, as an example from the convex hull of the data points in that bin.

One advantage of this approach is that the perturbation model is non-stationary w.r.t. the input space, since the perturbations are generated locally w.r.t. hashcode bins. Furthermore, the perturbation model does not make any parametric assumption about the global distribution of data or the data manifold. Mathematically, Eq. 8 can be re-expressed for the hashcodes based perturbation function as follows.

$$\bar{\mathbf{y}} \sim g(\mathbf{c}) \ s.t. \ \mathbf{h}(\mathbf{y}) = \mathbf{h}(\bar{\mathbf{y}}) = \mathbf{c}, \tag{9}$$

where, $\mathbf{h}(.)$ is an LSH function, represented by a set of $H$ hash functions, each outputting one bit, mapping an input $\mathbf{y}$ to its hashcode $\mathbf{c} \in \{0,1\}^H$. The perturbation model $g(.)$ samples a perturbation w.r.t. a hashcode bin, and not a single input. In practice, it is efficient to sample all the perturbations together across all the hashcode bins.

A pseudo code is presented in Alg. 1. The input data points $(\mathbf{y}^{(1)}, \cdots, \mathbf{y}^{(n)})$ are the inputs of a model that we regularize, i.e. a discriminator for our problem. First, we compute hashcodes for all the data points and arrange the data into their corresponding unique hashcode bins. For each bin, a Dirichlet distribution is then initialized with hyperparameter $\alpha$, and of dimension equal to the number of data points in the bin. For $n_i$ data points in the $i_{th}$ bin, we sample $n_i b$ number of perturbations from that bin. Each perturbation is sampled in-turn by randomly drawing a multinomial vector from the Dirichlet distribution, which acts as a random convex combination of all the points in the bin.

Since we are interested in local smoothness given by hashcodes bins representing local neighborhoods, we propose a regularization criterion based on information theory. In particular, we minimize the conditional entropy of the model outputs given the hashcode representation of the inputs:

$$\min_{f(.)} \mathcal{H}(f(\boldsymbol{\mathcal{Y}})|\boldsymbol{h}(\boldsymbol{\mathcal{Y}})). \tag{10}$$

Although $f(.)$ & $\mathbf{h}(.)$ are deterministic, both $f(\boldsymbol{\mathcal{Y}})$ & $h(\boldsymbol{\mathcal{Y}})$ are stochastic given their dependence on $\boldsymbol{\mathcal{Y}}$. Empirical estimate of this regularization term is easy and cheap to compute. For each hashcode bin, we compute the model outputs for existing as well as the sampled perturbations from Alg. 1, and then we compute an empirical estimate of entropy of

**Algorithm 1** Generate Perturbations via LSH

---

**Require:** $\{\mathbf{y}^{(1)}, \cdots, \mathbf{y}^{(n)}\}, \alpha, b$
1: $\mathbf{c}^{(1)}, \cdots, \mathbf{c}^{(n)} \leftarrow \text{computeHashcode}(\mathbf{y}^{(1)}, \cdots, \mathbf{y}^{(n)})$
2: $\mathbf{Y}^{(1)}, \cdots, \mathbf{Y}^{(m)} \leftarrow \text{hashcodeBin}(\{(\mathbf{y}^{(1)}, \mathbf{c}^{(1)})\}_{i=1}^n)$
   % $\mathbf{Y}^{(i)}$ *has all the inputs from* $i_{st}$ *hashcode bin*
3: **for** $i = 0 \rightarrow m$ **do**
4:     $n_i \leftarrow \text{countSamplesInBin}(\mathbf{Y}^{(i)})$
5:     $\bar{n}_i \leftarrow n_i * b$    % *no. of perturbations in* $i_{th}$ *bin*
6:     **for** $j = 0 \rightarrow \bar{n}_i$ **do**
7:         $\pi_j \sim Dir(\alpha \mathbf{1}_{n_i})$    % *sample convex combination*
8:         $\bar{\mathbf{y}}_j^{(i)} \leftarrow \pi_j * \mathbf{Y}^{(i)}$    % *perturbation in the bin*
9:     **end for**
10: **end for**
11: **Return** $\{(\bar{\mathbf{y}}_j^{(i)}, \mathbf{c}_j^{(i)})\}_{i=1, j=1}^{m, n_i}$

---

those outputs within the bin. Since the target variable is one-dimensional in our problem, i.e. observation $y_t$ for timestep $t$, computing entropy of the model outputs is easy even for the case of conditionals densities; one can, for example, use non-parametric estimators like histograms. This way, we iterate through all the bins to finally compute the conditional entropy term, $\mathcal{H}(f(\mathcal{Y})|h(\mathcal{Y}))$.

We use the regularization criterion in Eq. 10 in a non-invasive manner to either tune the vast space of hyperparameters of a model like GBM, or for data augmentation to regularize the model with LSH perturbations. As for tuning LSH model, both the type and number of hash functions can be tuned independently of the TE estimation problem. It's valuable to keep the mean (minimum) number of data points per bin above a certain threshold. The threshold values can be decided intuitively so as to impose the desired regularization upon the neural output function.

While our approach admits *any* LSH algorithm, we also propose a novel (greedy) algorithm for unsupervised learning of locality sensitive hash functions and use it for our experiments. See supplementary material for details.

## 3.2 THEORETICAL ANALYSIS

As discussed above, we use locality sensitive hashing (LSH) based regularization for learning the conditional log likelihood estimates by perturbing the inputs within the same hashcode bin. Perturbation may lead to a different distribution than the data distribution, and thus the conditional likelihood estimates derived from this distribution may be biased. We establish the conditions under which the perturbed distribution yields consistent estimates. Let $g_{n,H}(.)$ denote the histogram distribution obtained by using H locality sensitive hash functions and $n$ samples. We will see LSH based data generation as sampling from a data-driven histogram. Using this insight, results from Lugosi and Nobel [1996]

and a proof technique similar to Rothfuss et al. [2019], we demonstrate consistency of our sampling approach under some regularity conditions.

**Theorem 1.** *Let* $\lim_{n \to \infty} \frac{2^H}{n} \to 0$, $\lim_{n \to \infty} \frac{tH \log n}{n} \to 0$ *and the input space,* $\mathbf{y} \in \mathbb{R}^t$ *is bounded. Consider any function,* $f : \mathbf{y} \to (0, \infty)$ *with* $\log f$ *having finite second order moment w.r.t to* $p$ *and* $g_{n,H}$. *Then,*

$$\lim_{n \to \infty} |\mathbb{E}_p[-\log f(\mathbf{y})] - \mathbb{E}_g[-\log f(\mathbf{y})]| \to 0. \quad (11)$$

The above result establishes that the perturbed distribution will yield same estimates in expectation as $n$ becomes large. As a corollary, we can establish the consistency of our TE estimator. We can rewrite,

$$\mathcal{T}_{X \to Y}^q = \mathbb{E}_p \log \frac{q(y_t|\mathbf{y_{t-1}}, \mathbf{x_{t-1}})}{q(y_t|\mathbf{y_{t-1}})}, \quad (12)$$

and the TE estimated using $g_{n,H}$ be,

$$\mathcal{T}_{X \to Y}^{q,g} = \mathbb{E}_g \log \frac{q(y_t|\mathbf{y_{t-1}}, \mathbf{x_{t-1}})}{q(y_t|\mathbf{y_{t-1}})}. \quad (13)$$

**Corollary 1.** *If the conditions in Thm. 1 hold, and the model distribution,* $q > 0$ *everywhere. Then letting* $f(\mathbf{y_t}, \mathbf{x_{t-1}}) = \frac{q(y_t|\mathbf{y_{t-1}}, \mathbf{x_{t-1}})}{q(y_t|\mathbf{y_{t-1}})}$, *we have,*

$$\lim_{n \to \infty} |\mathcal{T}_{X \to Y}^q - \mathcal{T}_{X \to Y}^{q,g}| \to 0. \quad (14)$$

**Remark:** To find a good discriminator $q$, we optimize the LSH regularized MLE objectives, i.e., $\min_q -\mathbb{E}_{g_{n,H}} \log q(y_t|.)$. As $n$ becomes large, this is the same as computing expectation over data distribution $p$ due to Thm 1, $\min_q -\mathbb{E}_p \log q(y_t|.)$. If the function class is expressive enough, such as a neural network for which the universal approximation theorem holds, the optimal discriminator would correspond to the correct conditional distribution derived from population distribution [Lu and Lu, 2020].

The above results characterize the distribution of perturbed samples and the behaviour of estimates under that distribution. Another aspect of our approach is that it relies on finite sample size, and thus, we next characterize sample complexity of our estimator to obtain high-confidence estimates.

**Theorem 2.** *For some data distribution* $p$ *and conditional model distribution* $q$ *and* $-\log q(y_t|.) \in [-Q, Q]$. *Let* $\hat{\mathcal{T}}_{X \to Y}^q$ *denote the n-sample estimate of transfer entropy. Then with probability* $1 - \delta$ $(\delta > 0)$, *we have*

$$|\hat{\mathcal{T}}_{X \to Y}^q - \mathcal{T}_{X \to Y}^q| \leq 2Q\sqrt{(2/n) \ln(4/\delta)}. \quad (15)$$

As a consequence of the above result, we can bound the error variance as below:

$$\mathbb{E}(\hat{\mathcal{T}}_{X \to Y}^q - \mathcal{T}_{X \to Y}^q)^2 \leq 4Q^2((1-\delta)\frac{2}{n}\ln\frac{4}{\delta} + \delta). \quad (16)$$

The first term is the dominant term in above expression and thus, the variance of the estimator reduces at the rate of $\frac{1}{n}$.

## 4 EMPIRICAL EVALUATION

First, to demonstrate the efficacy of the proposed estimator when the ground truth is known, we evaluate on a synthetic dataset (Sec. 4.1). We also perform extensive analysis on two real world examples: a neuroscience dataset (Sec. 4.2) and a dataset representing US stock market activity (Sec. 4.3).

**Estimators for Comparison**  We evaluate four different baseline estimators of TE from the literature: (i) kNN estimator: *kNN*; (ii) Conditional kernel density estimation: *CKDE*; (iii) Copula entropy based estimator of Ma [2019], referred as *Copent*. (iv) Conditional mutual information based estimator of Zhang et al. [2019], referred as *ITENE*. To estimate TE in terms of conditional entropies directly using a discriminative model, we employ two neural models: Temporal Convolution Networks (TCN) and *N-Beats*, as well as Gradient Boosted Machines (GBM). LSH indicates that it is a perturbation model based on locality sensitive hashing (LSH): LSH-RC imposes regularization penalty for inconsistent model outputs, while LSH-A involves data augmentation. The GBM model uses LSH-RC for tuning a large set of hyper-parameters (GBM-LSH-RC*). For neural models, we use LSH-A (TCN-LSH-A* & NBeats-LSH-A*). For the baseline of Gaussian noise as a perturbation model, we likewise have GN-RC and GN-A. If no perturbation model is used, it is referred to as "No Reg.", another baseline. Standard regularization techniques like weight decay and dropout are used for all models, including "No Reg.". For continuous time series, the models are used as regressors trained with mean squared loss, and for discrete-valued time series as classifier with cross entropy loss. Each model has its own strength depending on the nature of data, so in some cases, we present results for the best of the three discriminative models (GBM, TCN, NBeats) accordingly.

**Parameter Settings**  In regards to tuning a discriminator (timeseries forecaster), hyperparameters are tuned for $q(y_t | \mathbf{y}_{t-1})$ alone, which is then used for $q(y_t | \mathbf{y}_{t-1}, \mathbf{x}_{t-1})$ as well. For instance, if we want to compute transfer entropy from every timeseries $j$ to a given timeseries $i$, we tune the hyperparameters of the discriminator only once, just using timeseries $i$. In the perturbation model, we generate new samples 10 times the original number of samples, i.e. b=10 in Alg. 1 (for LSH-A, $b = 3$). For sampling from Dirichlet distribution in Alg. 1, $\alpha$=0.1. Number of hash functions for LSH is 15, $H = 15$. These parameters do not require fine-tuning, so they were set manually. We explored various values of k for kNN estimator; k = 1, 3, 5 were equally good across all experiments, compared to a higher value of k.

### 4.1 EVALUATION ON SYNTHETIC DATA

Our algorithm for generating binary valued synthetic data is as follows. First we draw 3000 samples for the target

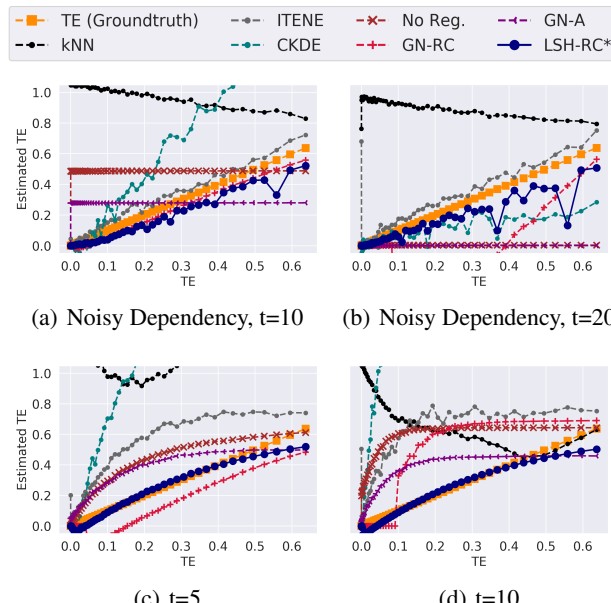

(a) Noisy Dependency, t=10   (b) Noisy Dependency, t=20

(c) t=5   (d) t=10

Figure 4: Estimates of TE are plotted w.r.t. the groundtruth for a synthetic dataset; $t$ refers to the time lag for how far back we look into the past to forecast for the current timestep. In 4(a) and 4(b), $y_t$ has dependence w.r.t. only one of the dimensions of $\mathbf{x}_{t-1}$ whereas in Fig. 4(c) and Fig. 4(d), all the dimensions of $\mathbf{x}_{t-1}$ have dependence w.r.t. $y_t$. The suffix "*" refers to the proposed TE estimator.

variable, $y_t$, s.t. $p(y_t = 1) = 0.5$. We assume zero temporal dependencies within time series Y, i.e. $H(\mathcal{Y}_t | \mathcal{Y}_{t-1}) = H(\mathcal{Y}_t) = \log(2)$. Next, we randomly select one of the $t$ dimensions of the conditioned variable $\mathbf{x}_{t-1}$, denoted by $x_r$, such that it depends on the target $y_t$, and the rest of the $t - 1$ dimensions are independent of $y_t$; $p(x_r = 1 | y_t = 1) = q, p(x_r = 1 | y_t = 0) = 1 - q$. This implies, $H(\mathcal{Y}_t | \mathcal{X}_{t-1}, \mathcal{Y}_{t-1}) = -q \log q - (1-q) \log(1-q)$, which is basically the entropy of a biased coin with probability of head equal to $q$. Overall, $\mathcal{T}_{X \to Y} = \log(2) - q \log q - (1 - q) \log(1 - q)$. For $q = 0$ or $q = 1$, $\mathcal{T}_{X \to Y}$ attains its maximum value of $\log(2)$. For $q = 0.5$, that is using an unbiased coin to sample $x_r$ given $y_t$, there is no transfer of entropy, $\mathcal{T}_{X \to Y} = 0$. We generate synthetic data for varying values of $q$, from $q = 0$ to $q = 0.5$. The most important aspect of this data generation step is that only *one* of the many dimensions in $\mathbf{x}_{t-1}$ is dependent on $y_t$, while the others are not. In Sec. 2, we referred to this as *noisy dependency*.

In Fig. 4, we present experimental results for the estimation of TE in the synthetic dataset.[1] Here, we only use GBM model as a discriminator. The figures differ by the dimension of the conditioned variables ($t = 20, t = 10, t = 5$), $\mathbf{x}_{t-1}$ or $\mathbf{y}_{t-1}$, i.e. how far back we look into the past for forecasting time series $Y$.

---

[1]Copent estimator is excluded from the figure since its estimates are way beyond the range of true TE.

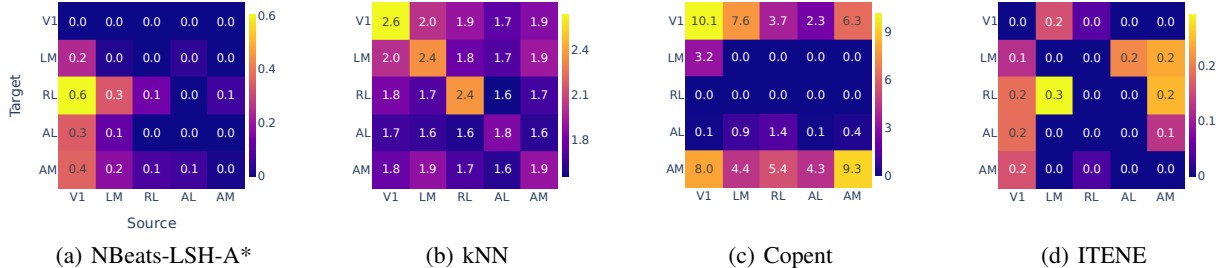

Figure 5: Estimates of TE between mouse visual areas. The matrices shows the TE values $\mathcal{T}_{\text{Source}\rightarrow\text{Target}}$ with source regions along columns and target regions along rows. The brain regions are sorted by ascending hierarchical order from left to right and top to bottom. V1 can be seen as the gateway of the visual system. The result of our method in (a) reveals the brain structure that matches current knowledge of the visual system, while others do not.

In reference to Fig. 4(a) and Fig. 4(b), the kNN-based estimator provides a severe overestimate of TE, and is almost agnostic to the dependency of $x_r$ on $y_t$, driven significantly by noisy signal from all the other $2t-1$ dimensions. Its error reduces when dependency between $x_r$ and $y_t$ approaches the highest value. Despite the popularity of the estimator, the results are unsurprising for the aforementioned technical reasons. CKDE estimates correlate to the ground truth values of TE, but with a significant error bias. ITENE obtains the TE estimates with very low error bias. Our approach of LSH-RC* also has very low estimation errors, although there are a few instances where it is high. One challenge is to optimize the trade-off between the log likelihood objective and the regularization term. In contrast, the baseline approaches of directly using a discriminator without regularization (No Reg.), and the Gaussian noise based data augmentation are both completely unsuccessful. Regularization using Gaussian noise based data perturbation model (GN-RC) seems to work for smaller time lag of $t = 10$.

Besides the above settings of noisy dependency, in Fig. 4(c) and Fig. 4(d), we present results for the case of $y_t$ being dependent w.r.t. all the dimensions of $\mathbf{x}_{t-1}$. In this setting, while our model obtains the best estimates, the baseline models perform relatively better than the former setting.

It is worth noting that the above described process of generating synthetic data is not artificial, but recreates highly noisy temporal dependencies observed between and within time series from domains such as neuroscience and finance.

## 4.2 EVALUATION ON NEUROSCIENCE DATA

Next, we applied the method to the neuroscience dataset, Allen Brain Observatory–Visual Coding Neuropixels [Siegle et al., 2021], by offering a metric from information theory perspective to discover the structure of the mouse brain and verify whether the results agree with the current findings of the visual system. Both anatomical and functional studies have shown that the brain visual system

is hierarchically organized and the visual information propagates across the cortical areas in order accordingly [Siegle et al., 2021, Harris et al., 2019]. During the early visual process, it is expected to see that the activities of low-order regions drive those of high-order regions.

Fig. 5 presents the results, showing estimated TEs between regions. The columns indicate the source regions and the rows indicate the target regions. The hierarchical order of the brain regions, from low to high, is V1, LM, RL, AL, and AM [Harris et al., 2019, Siegle et al., 2021], which are sorted along rows and columns. A larger value indicates the source region contributes more significantly to the target region's entropy, which implies the direction of information flow and the source region has an impact on the activity of the downstream target region. In Fig. 5(a), all large values concentrate in the bottom left triangle, which means the low-order regions impact the high-order regions, thus the conclusion agrees with the hierarchical order found by other anatomical or functional methods. In contrast, other methods in Fig. 5(b), (c), and (d) do not properly reveal the hierarchical relationships among the visual areas, especially they present many large positive TE values in the top right triangle matrix. For example, these methods show $\mathcal{T}_{\text{AM}\rightarrow\text{V1}} > 0$, meaning AM impacts V1, which is not reasonable bio-physically.

## 4.3 EVALUATION ON US STOCK DATA

We consider the top 64 most actively traded stocks in the US, and define two sets of time series for TE estimation. The first concerns the frequency of order arrivals during regular intervals: for a fixed interval of 100 milliseconds, we count the number of all orders for each stock over the course of 6 trading days. The second set of time series constitutes the observed mid-price (MP) changes over 1 second intervals for each of the same 64 stock, over a 14 day-period. In both settings, we consider context size of 120 timesteps to forecast the current timestep ($t = 120$).

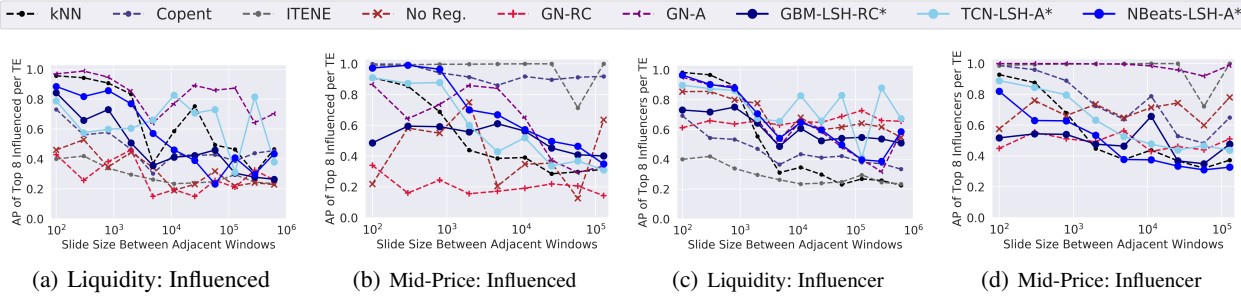

| (a) Liquidity: Influenced | (b) Mid-Price: Influenced | (c) Liquidity: Influencer | (d) Mid-Price: Influencer |

Figure 6: US Stocks: Average precision is computed as a measure of consistency between transfer entropy estimates from two adjacent time windows. The suffix "*" refers to the proposed TE estimators.

We use a window of 3000 timesteps (5 minutes) for estimating transfer entropy between each pair of 64 stocks; for the case of mid-price temporal dynamics, a window is of one hour (3600 timesteps). We construct many such windows of identical size during the period of 6 days (or 14 days), so that the time gap between two subsequent windows increases exponentially in time. We expect transfer entropy estimations from two windows close in time to be similar/consistent to each other, and significantly different, or inconsistent, as they are further apart in time (due to the inherent heteroskedasticity of competitive markets).

In this context, transfer entropy can be represented as a sequence of 64-by-64 matrices. We use two criteria for such an evaluation of consistency versus inconsistency. For a given security, we evaluate if the top 8 securities influenced by it are consistent between two adjacent windows (row-wise consistency). Denoting transfer entropy matrices from two adjacent windows as $T^{(1)}$ and $T^{(2)}$, we set the top 8 values in each row of $T^{(1)}$ as a positive label and the rest as negative labels, and then we use $T^{(2)}$ as scores w.r.t. the labels from $T^{(1)}$, so as to compute Average Precision (AP) of the top 8 influenced securities. We expect this score to drop as we increase the time gap between two adjacent windows, as the list of top securities influenced by a security should change as market conditions evolve.

In Fig. 6(a) and 6(b), we present the results for this evaluation criterion. For a slide size of up to 1000 timesteps between two adjacent windows, we expect a very high AP score (high consistency), since it is reasonably small compared to the window size of 3000 timesteps (or 3600 timesteps). As we increase the size to be a large multiple of the window size, average precision should drop, and then it may remain small or decrease further. NBeats-LSH-A* outperforms all other estimators, following the expected pattern of consistency for the criterion explained above. (We exclude CKDE from this evaluation since it doesn't scale to high dimensions, and performs poorly on the synthetic datasets.) Estimations with no regularization or Gaussian noise based perturbation (GN-RC & GN-A) lead to low consistency regardless of the slide size; in Fig. 6(a), GN-A

seems to provide high consistency for all the slide sizes which is not a desired pattern of consistency either. The consistency scores for the kNN estimator remain high for small slide sizes, but drops sharply, although sometimes it provides high consistency even at large slide sizes. Copent and ITENE estimators have consistency scores almost constant w.r.t. slide size, indicating their vulnerability to noise. The choice of the underlying discriminator in our estimation approach depends upon the phenomenon of interest; for instance, GBM-LSH-RC* performs well for modeling the order activity (discrete valued time series) whereas TCN-LSH-A* is better suited for modeling the temporal dynamics of mid-prices of stocks (continuous valued time series).

Another criterion for evaluation is to see if the top 8 securities influencing a given security are consistent between two adjacent windows. Similar to the previous exercise, we compute average precision but column-wise instead. Results shown in Fig. 6(c) and Fig. 6(d), exhibit similar patterns of superiority of our proposed estimator w.r.t. the baselines.

Overall, the experimental results suggest that transfer entropy estimation can be unreliable when dealing with long ranged and noisy temporal dependencies, as observed in real world domains like finance and neuroscience. Our proposed estimator, regularized using an LSH based perturbation model, shows robustness in the selected experiments.

## 5 CONCLUSIONS

We established that empirical estimation of transfer entropy between time series is a challenging problem, especially if temporal dependencies are long ranged and noisy. Such noise is common in domains such as finance and neuroscience, though difficult to characterize. We explained theoretical reasons for why well known estimators are prone to such noise, and propose a novel method - a discriminator regularized using a perturbation model based on locality sensitive hashing. We proved consistency of the estimator, and that it's variance decreases linearly in sample size. It is also shown to be efficient empirically in synthetic as well as real world settings of neuroscience and finance domain.

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
