# OpenReview forum: "Estimating Transfer Entropy under Long Ranged Dependencies"
_auai.org/UAI/2022/Conference — UAI 2022 Poster_

### Official Review · Reviewer_Qp5R · 2022-04-10

**Q2(1) Originality/Novelty:** 3
**Q2(2) Significance/Impact:** 2
**Q2(3) Correctness/Technical Quality:** 3
**Q2(6) Clarity Of Writing:** 3
**Q6 Overall Score:** 4
**Q8 Confidence In Your Score:** 4

**Q1 Summary And Contributions:**

The authors focus on the challenge of accurately estimating conditional entropies (CE), with the overall goal of estimating transfer entropies (differences in CEs). In high dimensional settings this is a difficult problem, so that authors propose a new estimator based on disciminative models trained using maximum likelihood.



**Q2 Assessment Of The Paper:**

More detailed information regarding each of these aspects is given below:

**Q2(4) Quality Of Experiments (Optional):**

3: Good: The experimental evaluation is adequate, and the results convincingly support the main claims.

**Q2(5) Reproducibility:**

2: Fair: Key resources (e.g., proofs, code, data) are unavailable but key details (e.g., proof sketches, experimental setup) are sufficiently well-described for an expert to confidently reproduce the main results.

**Q3 Main Strengths:**

The authors pose the challenge of conditional entropy estimation from the perspective of maximum likelihood discriminative models. The authors note that models trained to minimize ML objectives are implicitly estimating conditional likelihoods - the benefit that these methods enjoy is that they have been shown to to scale to higher dimensions. There are, however, a series of challenges associated with using discriminative models. To their credit the authors note on important limitation, which is the potential for such models to overfit, and propose and introduce a perturbation based regularization penalty. However, I think there are a series of other important issues that should be discussed/addressed (see below).

**Q4 Main Weakness:**

The overall idea is interesting and the authors have done a good job of considering potential issues associated with overfitting. However, in my opinion there are further issues around model mis-specification and model mis-calibration which are important to consider when using the output of over-parameterized neural network model as inputs to conditional entropy estimation (discussed in further detail below).

**Q5 Detailed Comments To The Authors:**

As mentioned above, the overall idea is interesting and the authors have done a good job of considering potential issues associated with overfitting. Another important and partially related issue is that of model mis-specification and mis-calibration which will affect the proposed method in the context of real valued and discrete valued time series respectively.

In the case of real valued time series, the main potential issue the proposed method could face in practice is model mis-specification. Concretely, equation (8) makes an assumption of homoscedasticity which is likely to be violated in practice. Essentially, the fixed variance in equation (8) implies that only the magnitude of residuals should be considered when computing the conditional entropies. If instead errors are heteroskedastic and this assumption is not satisfied, then all resulting conditional entropy calculations will be meaningless.

In the case of discrete valued time series, the authors note that they can use the cross-entropy loss as a measure of conditional entropy. Again, it is unclear whether this is a good idea when using over-parameterized neural networks which are widely reported to be poorly calibrated, sensitive to random initialization of weights and prone to overfitting.

Finally, I found the empirical comparisons difficult to follow and cluttered. It would have been more beneficial to have clear paired plots e.g., demonstrating the benefits of the proposed LSH perturbations.



**Q7 Justification For Your Score:**

The overall idea is interesting but has not been sufficiently explored to warrant acceptance at this stage.

**Q9 Complying With Reviewing Instructions:**

1: Yes.

---

### Official Review · Reviewer_Gp3Q · 2022-04-13

**Q2(1) Originality/Novelty:** 3
**Q2(2) Significance/Impact:** 2
**Q2(3) Correctness/Technical Quality:** 3
**Q2(6) Clarity Of Writing:** 4
**Q6 Overall Score:** 8
**Q8 Confidence In Your Score:** 3

**Q1 Summary And Contributions:**

In their paper, the authors highlight that in presence of long ranged and noisy temporal dependencies, the empirical estimation of transfer entropy between time series is challenging. To address this problem, the authors propose to express TE as a difference of two conditional entropy terms, which are estimated from conditional likelihoods computed in-sample from any  discriminative model that is trained by maximizing the conditional log likelihood of the target variable given the input variable

**Q10 Ethical Concerns (Optional):**

No Ethical Concerns

**Q2 Assessment Of The Paper:**

More detailed information regarding each of these aspects is given below:

**Q2(4) Quality Of Experiments (Optional):**

3: Good: The experimental evaluation is adequate, and the results convincingly support the main claims.

**Q2(5) Reproducibility:**

3: Good: Key resources (e.g., proofs, code, data) are available and key details (e.g., proofs, experimental setup) are sufficiently well-described for competent researchers to confidently reproduce the main results.

**Q3 Main Strengths:**

The authors’work is well motivated by an informative review of the state of the art.
The paper is pleasant to read.
The authors give a theoretical proof that (i) the proposed estimator is consistent under some mild regularity conditions, (ii) its variance decrease linearly as sample size increases.
The evaluation protocol includes experiments on both controled synthetic data and real-wold neuroscience dataset. The results show that the authors’ estimator outperforms 4 state-of-the-art estimators (kNN estimator, Conditional kernel density estimation, a Copula entropy based estimator, a Conditional mutual information based estimator of Zhang)

**Q4 Main Weakness:**

I can see none.

**Q5 Detailed Comments To The Authors:**

I have no detailed comment.

**Q7 Justification For Your Score:**

See Q3.

**Q9 Complying With Reviewing Instructions:**

1: Yes.

---

### Official Review · Reviewer_wrtS · 2022-04-14

**Q2(1) Originality/Novelty:** 2
**Q2(2) Significance/Impact:** 2
**Q2(3) Correctness/Technical Quality:** 3
**Q2(6) Clarity Of Writing:** 3
**Q6 Overall Score:** 7
**Q8 Confidence In Your Score:** 4

**Q1 Summary And Contributions:**

The authors propose a method to estimate transfer entropy between time series. The key contribution is the idea of using a generic discriminative model such as a neural network to output an estimation of transfer entropy, together with a procedure based on local sensitivity hashing to augment the training data points. The authors give theoretical results on the convergence of the estimator and prove that the method is effective through numerical experimentation.

**Q2 Assessment Of The Paper:**

More detailed information regarding each of these aspects is given below:

**Q2(4) Quality Of Experiments (Optional):**

3: Good: The experimental evaluation is adequate, and the results convincingly support the main claims.

**Q2(5) Reproducibility:**

2: Fair: Key resources (e.g., proofs, code, data) are unavailable but key details (e.g., proof sketches, experimental setup) are sufficiently well-described for an expert to confidently reproduce the main results.

**Q3 Main Strengths:**

The paper is overall well written, despite some minor typos. The presentation is clear, the state of the art is well documented, and the main mathematical claims are proved. And the asymptotic correctness of the proposed estimator is proved mathematically.
The method proposed looks interesting and it is substantiated by numerical experiments.


**Q4 Main Weakness:**

I believe that the main weakness is that, despite the proof that asymptotically (as the number of samples increases to infinity) the estimator converges to the true value of transfer entropy, it is less clear how to make it work from a practical perspective: there are a number of hyperparameters to be gauged (the number of LHS functions, the type of LHS functions, the type of neural network) and it is not clear how they affect the estimator.

It is not clear how, practically, one could choose the hash function to augment the input data, and how the dimensionality of the output of the hash function impacts the results (the number of LHS functions, the type of LHS functions, the type of neural network). It seems that the only way to do it is by trial and error.

Some technical details and explanations are missing in the description of how the model is trained, please see more details in the sequel.

**Q5 Detailed Comments To The Authors:**

In general, the argument are clearly exposed, but when the discussion becomes more technical and rigorous notation is required, then it becomes more difficult to follow. Notation should be reviewed.

It is not clear how the model is trained: in section 3, it is written that “maximizing the conditional log-likelihood of the target variable given the input variable can be employed as an estimator of conditional entropy. For discrete-valued time series with support set C, a classifier trained by cross-entropy loss can provide an empirical estimate of conditional entropy itself”, therefore the idea is to maximize a conditional log-likelihood. Then, in section 3.1, equation 12, a regularization term is proposed, to handle the new data points added to the training dataset. At this point, it is still not clear how this term enters the optimization. Finally, at the remark under corollary 1 an expression of the function to be minimized is given. This expression is not clear enough to me: in my understanding q should be the model to be fitted. I do not understand why y_t enters q as a variable if it should be the output of the model. Moreover, the augmented points cannot have a label y_t, therefore this notation cannot apply to those points. The notation here is ambiguous and, even though I believe I understood what the authors propose to optimize, I cannot say I am completely certain. This remark has to be thoroughly rewritten in order to make transparent to the reader how the underlying optimization exactly works so that this approach could be reproduced.


As a general comment, theorems and propositions should be as self-contained as possible, in the sense that all the symbols used to state the proposition should be defined within the theorem itself, otherwise it becomes hard for the reader to look for the meaning of the symbol in the preceding paragraphs: as an example, in theorem 1 it would be better to first define H, n, t, p and g, and then give the hypothesis and the thesis. Also, in the same theorem, it is stated that the input-space of function f, namely y, is bounded, and it is stated that r\inR^t. If y is a subset of R^t, than the symbol \subset should be used, because y is not an element of R^t, rather a subset. Generally, capital letters are used for subsets of vector spaces.

In equation (7) it took me some time to understand that the symbol [c] means the c-th entry of a histogram modeling the conditional probability function over a discrete support C. At equation (7) the authors denote such log-likelihood as log f(y_{t-1}^i)[c]. In the line following the equation, the same quantity is denoted as f_c(y^i_{t-1}). I believe the two symbols have the same meaning, but switching between different notations creates confusion and it is a big burden for the reader.

At equation (9) the same symbol, ‘q’, is used to denote two different ML estimators.

From equation (11) and the following discussion, I infer that the authors believe that convex combinations of points mapped to the same hash code are still mapped to the same hash code. Is this a trivial fact? I suspect this is not an obvious fact. The authors should clarify this point.

On a minor note:
In the abstract ‘copula’ is used with a capital ‘c’, I believe it should be rewritten as ‘copula’
In the introduction, transfer entropy is presented as a ‘relatively new’ concept, and a reference from 2000 is given. One would say that a concept defined 22 years ago is a well-established concept, rather than a relatively new one.

In multiple points I read the sentence "estimates are not overfit to the data", It sounds more natural to me to state that "estimates do not overfit the data".

Low-signal-to-noise ratio should be rewritten as Low signal-to-noise ratio.

In equation 5 in the last mutual information term there is a typo: ‘|’ should be substituted with ‘:’, otherwise I cannot understand it

On page 5, after equation (7), it is written that “it is well known that a proposal distribution q(.) for p(.) upper bounds the corresponding entropy function” ‘bounds from above” sounds better than “upper bounds”

At the bottom of page 8, “it’s variance” should be rewritten as “its variance”

I suggest thorough proofreading as there may be more typos.

**Q7 Justification For Your Score:**

I believe this is an interesting contribution: the authors propose a method to estimate transfer entropy and they prove mathematically that the estimator is asymptotically correct. Moreover, they prove experimentally that their method is effective.

**Q9 Complying With Reviewing Instructions:**

1: Yes.

---

### Decision · Program_Chairs · 2022-05-15

**Decision:**

Accept (Poster)

**Comment:**

Meta Review: The authors introduce a novel method to estimate transfer entropy between long-ranged dependency time series with the key novelty of using a generic discriminative model together with a local sensitivity hashing to augment the training data. The authors show theoretical convergence results and numerical results underpinning its advantages.

Pros:
* Original idea of transferring methodological knowledge from one domain to another
* Mathematical proofs of asymptotic convergence
* Good clarity and well-written, informative review of the state of the art

Cons:
* Not much info on how to tune the neural net in practical applications
* Unclear how estimator performs against other estimators in non-long-range situations

The reviewer's vote (confidence) was 7(4) / 8(3) / 4(4). The last reviewer's concern was that model-misspecification could pose a serious problem. To this the authors replied that the estimator allows to choose from a plethora of methods (informed by domain knowledge) to accommodate issues such as heteroskedasticity. This to me is subsumed under the con that, as any neural net, domain expertise is needed, and to me this is not a strong argument against the paper. Rather, future research within application domains can address this. Hence, accept.

Requirement for camera-ready version:
Next to the points that the authors promised to address, I would like to see numerical results showing how the estimator works for varying sample sizes in comparison to the other baselines and also for dependencies with and without the long-range property. Such an analysis would be needed for users to assess when this estimator is useful.